# Endothelin-1 and LOX-1 as Markers of Endothelial Dysfunction in Obstructive Sleep Apnea Patients

**DOI:** 10.3390/ijerph18031319

**Published:** 2021-02-01

**Authors:** Monika Kosacka, Anna Brzecka

**Affiliations:** Department of Pulmonology and Lung Oncology, Wroclaw Medical University, ul. Grabiszynska 105, 53-439 Wroclaw, Poland; anna.brzecka@umed.wroc.pl

**Keywords:** LOX-1, endothelin-1, sleep apnea, endothelial dysfunction, cardiovascular diseases

## Abstract

Introduction: The search of biochemical markers of endothelial dysfunction: lectin-like oxidized low-density lipoprotein receptor-1 (LOX-1)—involved in atherosclerotic plaques formation—and endothelin-1 (ET-1)—potent vasoconstrictor-might help in detecting obstructive sleep apnea (OSA) patients at high risk of cardiovascular diseases. Material and Methods: In 71 OSA patients (apnoea/hypopnoea index, AHI 28.2 ± 17.9/hour) and in 21 healthy controls the serum levels of LOX-1 and ET-1 were measured. Results: There were increased levels of ET-1 (1.58 ± 0.65 vs. 1.09 ± 0.38 pg/mL; *p* < 0.001) but not of LOX-1 in OSA patients as compared with healthy controls. In the patients’ group ET-1 levels negatively correlated with serum LDL levels. LOX-1 levels positively correlated with fasting glucose levels and were higher in the patients with than without diabetes. Neither ET-1 nor LOX-1 correlated with OSA severity. In mild OSA patients, there was a negative correlation between LOX-1 and mean arterial oxygen saturation during sleep. In severe OSA patients, there was a positive correlation between LOX-1 levels and uric acid. Conclusion: There is endothelial dysfunction in OSA patients as indicated by increased serum levels of ET-1 and possibly endothelial dysfunction in diabetic OSA patients as indicated by increased serum levels of LOX-1 and its correlation with fasting glucose levels.

## 1. Introduction

Obstructive sleep apnea (OSA) is a common disorder characterized by repetitive episodes of partial or complete obstruction of the upper airway during sleep, which cause recurring oxygen desaturation and arousal during sleep [1,2,3] This leads to many consequences, among which most importantly intermittent hypoxia and sleep fragmentation [4]. In OSA patients, there is a high incidence of diabetes mellitus and cardiovascular diseases (CVD), especially coronary heart disease, systemic hypertension, heart failure, cardiac arrhythmias, stroke, and diabetes mellitus [2,5]. Many mechanisms are involved in increased cardiovascular risk. The most important are chronic activation of the sympathetic nervous system, oxidative stress, chronic inflammation, and endothelial dysfunction [6,7,8,9,10]. In OSA patients higher expression of adhesion molecules was described in leukocytes, platelets, monocytes, and endothelial cells. In addition, monocytes and lymphocytes express increased avidity to endothelial cells [11]. Increased levels of adhesion molecules were also confirmed in the circulation. In the meta-analysis, numerous markers as CRP, TNF-alfa, IL-6, IL-8, ICAM (intracellular adhesion molecule), VCAM (vascular cell adhesion molecule) and selectins, were demonstrated to be higher in OSA patients as compared to control subjects [8].

Lectin-like oxidized low-density lipoprotein receptor-1 (LOX-1) belongs to type II glycoproteins. This 50-kDa transmembrane glycoprotein is a cell surface receptor for oxidized low-density lipoprotein (ox-LDL) [12,13]. It is expressed in different cells, including endothelial cells, vascular smooth muscle cells, macrophages, platelets, cardiomyocytes, and fibroblasts [13,14]. LOX-1 can be cleaved and released in a soluble form (sLOX-1) in the circulation [15].

LOX-1 plays an important role in atherosclerosis and vascular inflammation. It is treated as a mediator of endothelial dysfunction [16,17]. LOX-1 was found to be involved in endothelial injury, leukocyte recruitment, foam cell formation, and plaque rupture [18]. The interaction between ox-LDL and LOX-1 stimulates the production of various adhesion molecules such as VCAM-1 and cytokines, including MCP-1. This leads to the increased attachment and migration of inflammatory cells to the intima [13].

Endothelin-1 (ET-1) is a 21-amino acid peptide, which together with two other isoforms ET-2 and ET-3, belongs to the endothelin family. ET-1 is the most common type seen in humans [19]. ET-1 is mostly produced by endothelial cells but could be also secreted by epithelial cells, macrophages, smooth muscle cells, and fibroblasts. ET-1 is known as an extremely potent vasoconstrictor [20]. ET-1 is also involved in many pathophysiological processes, such as vascular oxidative stress, inflammation, reduced nitric oxide bioavailability, and impaired endothelium-dependent dilatation, and for this reason, it contributes to the development and progression of atherosclerosis [21,22]. The role of ET-1 has been widely described in many diseases, especially in coronary artery disease, myocardial infarction, heart failure, and hypertension. Taking into account the predictor and prognostic role of ET-1 in many cardiological diseases, the majority of authors indicate that, ET-1 should be treated as a potential risk marker for cardiovascular events [23].

As mentioned before, both LOX-1 and ET-1 are involved in the inflammatory process and contribute to endothelial dysfunction. In addition, LOX-1 stimulates ET-1 expression in human endothelial cells [24].

A better understanding of the molecular mechanisms involved in endothelial dysfunction is crucial for the identification of OSA patients with increased cardiovascular risk and searching in the future for new targets and strategies to prevent cardiovascular diseases in OSA patients. Therefore we undertook a study to investigate the serum levels of LOX-1 and ET-1 in OSA patients and correlations between these markers and increased risk of cardiovascular complications in OSA.

## 2. Materials and Methods

### 2.1. Patients and Control Subjects

A total of 74 patients with newly diagnosed OSA were enrolled in the study. The mean age of the patients was 55.58 ± 11.03 years and the mean apnoea/hypopnoea index (AHI) was 28.20 ± 17.92/hour. The examined group comprised 51 males and 23 females. Mean body mass index (BMI) was 32.68 ± 7.05 kg/m^2^. There were 20 patients with mild OSA (AHI 5–15/hour, mean 9.52 ± 2.69/hour), 24 with moderate OSA (AHI 15–30/hour, mean 20.19 ± 4.53/hour), and 30 with severe OSA (AHI > 30/hour, mean 47.06 ± 11.20/hour). The following cardiovascular diseases coexisted with OSA: hypertension in 50 patients, ischemic heart disease in 19, diabetes in 14, hypercholesterolemia in 23, and three patients underwent stroke. All the patients received standard treatment for co-morbidities. There were no differences in AHI between OSA patients with and without diabetes (33.7 ± 19.12/hour vs. 26.81 ± 17.66; *p* = 0.216), however, DI was higher and the mean arterial oxygen saturation (SaO_2_) was lower (90.75 ± 3.42 % vs. 91.74 ± 8.81%; *p* = 0.025) in the OSA patients with than without diabetes (29.26 ± 17.94/hour vs. 21.70 ± 8.20; *p* = 0.035).

The control group consisted of 21 healthy subjects, including 12 females, without any diseases, and especially without diabetes. The mean age was 52.80 ± 14.19 years, the mean AHI was 2.15 ± 1.82/hour and the mean BMI was 29.78 ± 7.15 kg/m^2^ in this group.

In the clinical examination of upper airways in the patients and the controls, no important narrowing of the nose or pharynx was found.

### 2.2. Polysomnography

All the patients and all subjects from the control group underwent nocturnal polysomnography using the Alice 6 LDe Polysomnographic Sleep System (Philips Respironics). During 8 h of nocturnal sleep, we measured: airflow with the use of an oronasal thermal sensor and nasal pressure sensor, chest and abdomen movements, body position, snoring, oxygen saturation using a finger clip sensor, and sleep stages. According to the standard criteria of the American Academy of Sleep Medicine (AASM): apnea was defined as a drop in the peak signal excursion by ≥90% of pre-event baseline for more than 10 s and hypopnea as a reduction in the airflow by at least 30% of pre-event baseline using nasal pressure in association with either ≥3% arterial oxygen desaturation or arousal [25]. In all cases, manual scoring was carried out after automatic scoring. In the present study, sleep stages were not analyzed separately, but parameters connected with sleep disturbances were observed during sleep stages. The following parameters were used in OSA diagnosis and severity assessment: AHI, oxygen desaturation index-ODI, mean SaO_2_ during sleep, and minimum SaO_2_ at the end of sleep apnoea/hypopnoea episodes.

### 2.3. LOX-1 and ET-1

Samples of venous blood were taken in the morning after fasting overnight. After centrifugation for 10 min at 1467 RCF, the serum was extracted and stored at −80 °C until examination. ET-1 and serum LOX-1 levels were measured using the enzyme-linked immunosorbent assay (ELISA) method with R&D Systems, Minneapolis, USA for ET-1 and MyBioSource for LOX-1. The tests were performed according to the manufacturer’s specifications. The ELISA microplate reader from MRXe Dynex Technologies (Chantilly, VA, USA) was used.

The following biochemical parameters were measured in the blood serum sample: CRP, total cholesterol, LDL cholesterol, HDL cholesterol, triglycerides, and uric acid.

### 2.4. Statistical Analysis

Statistical analysis was performed using the CSS Statistica software for Windows (version 5.0). Data were presented as means ± SD. Mean values between the two groups were compared by the Mann–Whitney U test and Spearman’s *r* correlation coefficient was used to assess the relationship between the two variables. Differences between samples were considered significant at *p* < 0.05.

### 2.5. Approval of Commission of Bioethics

This work has been approved by the Commission of Bioethics at Wroclaw Medical University (Approval No 412/2018). Written informed consent from all participants involved in the study was obtained.

## 3. Results

The levels of LOX-1 in OSA patients and healthy controls are similar (Table 1). In OSA patients LOX-1 levels positively correlated with fasting glucose levels (r_s_ = 0.23; *p* = 0.042). LOX-1 levels were higher in OSA patients with diabetes than in OSA patients without diabetes (1527.12 ± 901.69 pg/mL vs. 1024.24 ± 530.66 pg/mL; *p* = 0.03) (Figure 1). There was also a tendency for higher LOX-1 levels in diabetic OSA patients than in the control group (1527.12 ± 901.69 vs. 983.62 ± 615.18 pg/mL; *p* = 0.06). In severe OSA patients, there was a positive correlation between LOX-1 levels and uric acid (r_s_ = 0.38; *p* = 0.037). In mild OSA patients, there was a negative correlation between LOX-1 and mean SaO_2_ during sleep (r_s_ = −0.47; *p* = 0.03), although, in the whole OSA group, there were no correlations between LOX-1, neither in the sleep breathing disturbance parameters nor in the biochemical parameters (Table 2).

There were increased levels of ET-1 (1.58 ± 0.65 pg/mL vs. 1.09 ± 0.38 pg/mL; *p* < 0.001) in OSA patients as compared with healthy controls (Figure 2). In the patient group, ET-1 levels negatively correlated with serum LDL levels (r_s_ = −0.26; *p* = 0.027), otherwise, there was no correlation between ET-1 levels and the other biochemical parameters or the parameters of sleep breathing disturbances (Table 2).

The correlation between ET-1 and LOX-1 was not observed (r_s_ = 0.06; *p* = 0.603).

## 4. Discussion

In our study, LOX-1 levels were not significantly different in OSA patients than in healthy controls but were higher in OSA patients with diabetes than without and positively correlated with fasting glucose levels. Additionally, a negative correlation between LOX-1 and mean SaO_2_ during sleep was found, but only in mild OSA patients.

LOX-1 levels have been rarely studied in OSA patients. In one study, LOX-1 levels were significantly higher in OSA patients compared to the controls and correlated with OSA severity [26]. In this study, however, the examined group differed from our patients, as they comprised less obese patients and without ischemic heart disease or diabetes. To the best of our knowledge, the association between LOX-1 concentration in diabetic and non-diabetic OSA patients has not been studied. In the group of patients with unknown OSA incidence, increased levels of LOX-1 by 9% were previously found in diabetic patients compared to non-diabetic patients [27]. In this study, reduction of LOX-1 levels correlated with glycemic control and with the improvement in hemoglobin A1c; in our patients, diabetes was controlled, although the concentrations of hemoglobin A1c were not measured. In vitro studies show that glucose and glycoxidized LDL are associated with increased LOX-1 expression and LOX-1 production [28,29].

In our OSA patients, there was no correlation between LOX-1 levels and the incidence of hypertension, ischaemic heart disease, or stroke. In the non-OSA patients, elevated LOX-1 levels were found in the patients with hypertension, and especially with ischemic heart disease [30,31]. In the patients with ischemic heart disease, it was postulated that its levels could be helpful in predicting disease progression and a higher risk of future cardiovascular events [32]. In one multicenter study, the relationship between LOX-1 and long-term cardiovascular outcomes in patients undergoing percutaneous coronary intervention was found; patients with increased LOX-1 levels had a significantly higher incidence of major adverse cardiovascular and cerebrovascular events [32].

In our study, there was a positive correlation between LOX-1 and uric acid in patients with severe OSA. To the best of our knowledge, this association in OSA patients has not been studied previously. The correlation between LOX-1 and uric acid was described in healthy men [33] and in patients with coronary artery disease [34]. In vitro study show, that xanthine oxidase, which catalyzes the oxidation of xanthine to uric acid, stimulates LOX-1 expression on macrophages and vascular smooth muscle cells and induces foam cell formation through LOX-1 [35]. The role of uric acid in OSA is well described [36]. In our previous study of OSA patients, we showed that uric acid is negatively correlated with the mean and minimal SaO_2_ and positively with the ODI, and patients with hyperuricemia has a higher prevalence of hypertension and ischemic heart disease [37]

The results of our study confirm previously described increased ET-1 levels in OSA patients [38,39,40], although in one early study there were no differences in ET-1 levels in OSA patients and controls [41]. In some studies, a correlation between OSA severity and ET-1 level was found [42]. The ET-1 levels in OSA patients decrease after treatment with continuous positive airway pressure, as shown by a recent meta-analysis encompassing 375 patients [43]. As in the other clinical situations, ET-1 level in OSA patients is associated with increased cardiovascular risk; it was implicated in the rise of blood pressure observed during obstructive sleep apnoeas/hypopnoeas [44].

There was a surprising inverse correlation between serum ET-1 and LDL cholesterol levels in our patients. However, the interpretation of LDL levels could be difficult in the present study, because some of our patients received statins. LDL cholesterol has known atherogenic features [45] and ET-1 is a peptide also involved in atherosclerosis development [46]. Thus one could expect that in the patients with both of these molecules elevated, the incidence of cardiovascular diseases would be higher. Increased LDL cholesterol levels are usually observed in OSA patients [47] and correlate with OSA severity, as assessed by AHI calculated for non-rapid-eye-movement sleep [48]. We did not find studies investigating reciprocal relationships between ET-1 and LDL cholesterol in OSA patients.

*Limitations of the study.* Although we compared the groups of patients with and without diabetes, the concentrations of hemoglobin A1c were not measured and thus diabetes implications were only partially studied.

*Future perspective.* The influence of preventing upper airway obstruction episodes during sleep on the endothelial function in OSA patients should be the aim of further studies, especially, in groups of OSA patients with different severities of the syndrome and with and without diabetes.

## 5. Conclusions

The study confirms endothelial dysfunction in OSA patients as indicated by increased serum levels of ET-1 and possibly endothelial dysfunction in diabetic OSA patients as indicated by increased serum levels of LOX-1 and their correlation with fasting glucose levels.

## Figures and Tables

**Figure 1 ijerph-18-01319-f001:**
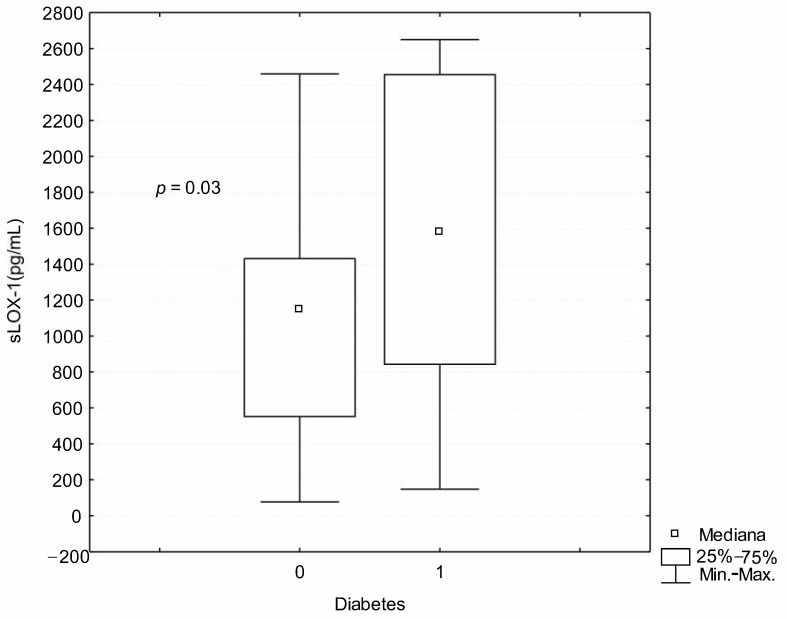
The comparison of LOX-1 serum levels in OSA patients with and without diabetes.

**Figure 2 ijerph-18-01319-f002:**
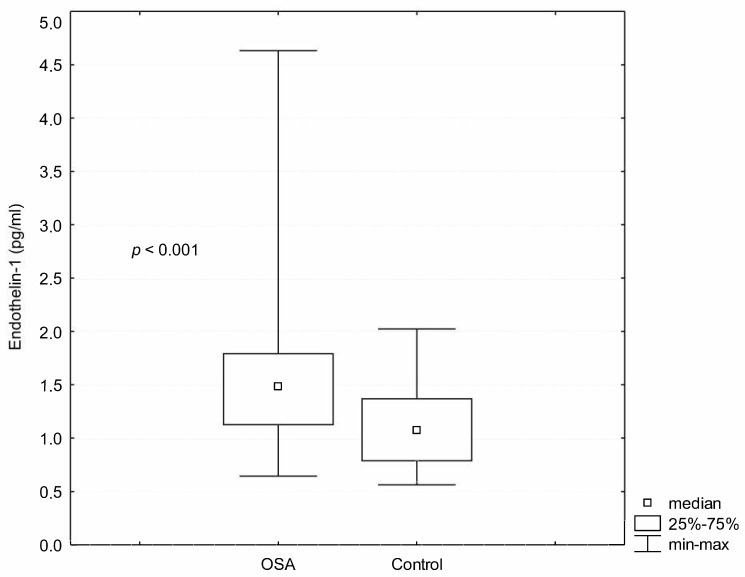
The comparison of Endothelin-1 serum levels in OSA patients and in the control group.

**Table 1 ijerph-18-01319-t001:** The comparison of OSA patients with healthy subjects from the control group.

Parameters	OSA Patients	Control Group	*p*
Age (years)	55.58 ± 11.03	52.80 ± 14.19	0.55
BMI (kg/m^2^)	32.63 ± 7.05	29.78 ± 7.15	0.07
LOX-1 (pg/mL)	1127.50 ± 642.01	983.62 ± 615.18	0.435
ET-1 (pg/mL)	1.58 ± 0.65	1.09 ± 0.38	<0.001
AHI (/hour)	28.20 ± 17.90	2.15 ± 1.82	<0.00001
DI (/hour)	23.75 ± 19.12	2.81 ± 2.32	<0.00001
Mean SaO_2_ during sleep (%)	91.37 ± 8.19	94.05 ± 2.07	0.006
Minimum saturation (%)	79.46 ± 8.83	86.05 ± 4.90	0.0006
Glucose (mg/dL)	95.96 ± 14.98	89.50 ± 8.56	0.037
CRP (mg/L)	4.02 ± 3.20	3.13 ± 2.96	0.437
Urid acid (mg/dL)	5.90 ± 1.50	4.90 ± 1.05	0.006
Total cholesterol (mg/dL)	203.23 ± 38.66	218.00 ± 52.87	0.197
LDL cholesterol (mg/dL)	122.21 ± 32.58	136.73 ± 45.09	0.210
HDL cholesterol (mg/dL)	50.49 ± 12.91	51.15 ± 14.81	0.377
Triglycerides (mg/dL)	156.36 ± 87.58	150.21 ± 73.74	0.942

**Table 2 ijerph-18-01319-t002:** The correlations between selected parameters and serum LOX-1 and ET-1 levels in OSA patients.

Parameters	LOX-1 (pg/mL)		ET-1 (pg/mL)	
	R Spearman	*p*	R Spearman	*p*
Age (years)	0.08	0.45	0.079	0.499
BMI(kg/m^2^)	0.10	0.38	−0.032	0.785
AHI (/hour)	0.04	0.71	−0.036	0.757
DI(/hour)	0.03	0.78	−0.102	0.386
Mean SaO_2_ during sleep (%)	−0.20	0.08	−0.097	0.411
Minimum SaO_2_ during sleep (%)	0.02	0.82	0.076	0.514
Glucose (mg/dL)	0.23	0.04	0.063	0.596
CRP (mg/L)	−0.05	0.63	0.028	0.809
Urid acid (mg/dL)	0.17	0.15	−0.001	0.989
Total cholesterol (mg/dL)	0.08	0.47	−0.203	0.083
LDL cholesterol (mg/dL)	−0.02	0.85	−0.263	0.027
HDL cholesterol (mg/dL)	0.009	0.93	0.163	0.167
Triglicerides (mg/dL)	0.18	0.10	−0.098	0.411

## Data Availability

Not applicable.

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
