# Peer review of "Endothelin-1 and LOX-1 as Markers of Endothelial Dysfunction in Obstructive Sleep Apnea Patients"

_ijerph, 2021, doi:10.3390/ijerph18031319_

Round 1

Reviewer 1 Report

This is an original article that evaluates the endothelial dysfunction in OSA patients studying serum levels of ET-1 and LOX-1. It is a really interesting work. However, I have some doubts.

Materials and methods:

In “patients” section it is not described the grade of OSA disease in patients affected by diabetes. Moreover, there is no report if any patients of the control group presented diabetes or other diseases.

Results and discussion

“In OSA patients LOX-1 levels positively correlated with fasting glucose levels (rs=0.23; p=0.042) and were higher in the patients with than without diabetes (1527.12±901.69 pg/ml vs 1024.24±530.66 pg/ml; p=0.03)”

Could be this dependent on diabetes despite OSA disease? As reported in literature LOX-1 could be associate with diabetes. Therefore, the lack of comparison of LOX 1 levels in OSA patients with and without diabetes should be considered as a limitation of the study, especially because it was affirmed that diabetes was controlled, but the concentrations of hemoglobin A1c were not measured.

Conclusion

According to previous observations, I think that is not correct to affirm that “There is endothelial dysfunction … in diabetic OSA patients as indicated by in-creased serum levels of LOX-1” because diabetes implications were partially studied.

Reviewer 2 Report

I appreciated the authors' approach to a very interesting and topical topic but in my opinion this interesting study requires further investigations regarding the causes of OSA in selected patients assessed by:

  • ENT examination
  • rhinofibrolaringoscopy with Muller maneuver 
  • sleep endoscopy

Secondly the impact of treatment (medical, surgical or odontostomatological) on the variation of inflammation markers considered (LOX-1 and ET-1) should be accurately assessed before and after a few months from the tailored therapy.

This approach could add originality to a study in which the reported data are largely widely known.

Round 2

Reviewer 1 Report

The authors answered all my doubts and resolved them.

I think that the article is original and good written, therefore it deserves to be published

Reviewer 2 Report

The absence of narrowing of the upper airways in the selected patients clarifies my perplexities indicated in the previous review. In my opinion the work can be published